**Data Availability Statement:** All relevant data are within the paper and its Supporting Information files.

# Perceptions of self-monitoring dietary intake according to a plate-based approach: A qualitative study

Maryam Kheirmandparizi[1], Jean-Philippe Gouin[2,3], Celeste C. Bouchaud[2], Maryam Kebbe[4], Coralie Bergeron[1,2], Rana Madani Civi[1], Ryan E. Rhodes[5], Biagina-Carla Farnesi[6], Nizar Bouguila[7], Annalijn I. Conklin[8], Scott A. Lear[9], Tamara R. Cohen[1,2]*

1 Faculty of Land and Food Systems, Food, Nutrition and Health, the University of British Columbia, Vancouver, British Columbia, Canada, 2 PERFORM Centre, Concordia University, Montreal, Quebec, Canada, 3 Department of Psychology, Concordia University, Montreal, Quebec, Canada, 4 Faculty of Kinesiology, University of New Brunswick, Fredericton, New Brunswick, Canada, 5 School of Exercise Science, Physical & Health Education, University of Victoria, Victoria, British Columbia, Canada, 6 Division of Adolescent Medicine, Montreal Children's Hospital, Westmount, Quebec, Canada, 7 Concordia Institute for Information Systems Engineering, Engineering, Computer Science and Visual Arts Integrated Complex, Concordia University, Montreal, Quebec, Canada, 8 Faculty of Pharmaceutical Sciences, the University of British Columbia, Vancouver, British Columbia, Canada, 9 Faculty of Health Sciences, Burnaby and Division of Cardiology, Providence Health Care, Simon Fraser University, Vancouver, BC, Canada

* tamara.cohen@ubc.ca

## Abstract

Dietary self-monitoring is a behaviour change technique used to help elicit and sustain dietary changes over time. Current dietary self-monitoring tools focus primarily on itemizing foods and counting calories, which can be complex, time-intensive, and dependent on health literacy. Further, there are no dietary self-monitoring tools that conform to the plate-based approach of the 2019 Canada Food Guide (CFG), wherein the recommended proportions of three food groups are visually represented on a plate without specifying daily servings or portion sizes. This paper explored the perceptions of end-users (i.e., general public) and Registered Dietitians of iCANPlate™—a dietary self-monitoring mobile application resembling the CFG. Qualitative data were collected through virtual focus groups. Focus group questions were based on the Capability, Opportunity, Motivation-Behaviour (COM-B) theoretical framework to explore perceptions of using the CFG and currently available dietary self-monitoring tools. The prototype iCANPlate™ (version 0.1) was presented to gain feedback on perceived barriers and facilitators of its use. Focus group discussions were audio recorded and verbatim transcribed. Trained researchers used thematic analysis to code and analyze the transcripts independently. Seven focus groups were conducted with Registered Dietitians (n = 44) and nine focus groups with members from the general public (n = 52). During the focus groups, participants mainly discussed the capabilities and opportunities required to use the current iteration of iCANPlate™. Participants liked the simplicity of the application and its capacity to foster self-awareness of dietary behaviours rather than weight control or calorie counting. However, concerns were raised regarding iCANPlate™'s potential to improve adherence to dietary self-monitoring due to specific characteristics (i.e.,

**Funding:** This study was funded by Social Sciences and Humanities Research Council (TRC), Grant number: SSHRC: 430-2020-00235 and the R. Howard Webster Foundation through Concordia University (TRC). The funders had no role in study design, data collection and analysis, decision to publish, or preparation of the manuscript.

**Competing interests:** None to declare.

**Abbreviations:** CFG, Canada's food guide; COM-B, Capabilities, Opportunities, and Behaviour; RD, Registered Dietitian; COREQ, Consolidated Criteria of Reporting Qualitative Research.

insufficient classifications, difficulty in conceptualizing proportions, and lack of inclusivity). Overall, participants liked the simplicity of iCANPlate™ and its ability to promote self-awareness of dietary intakes, primarily through visual representation of foods on a plate as opposed to reliance on numerical values or serving sizes, were benefits of using the app. Findings from this study will be used to further develop the app with the goal of increasing adherence to plate-based dietary approaches.

## Introduction

As a behaviour change technique, dietary self-monitoring (e.g., diet tracking) plays a key role in adopting and maintaining new dietary behaviours [1]. Self-monitoring functions by increasing self-awareness of one's actions and the conditions under which they occur [2, 3]. According to Control Theory [1], despite an individual's intentions to change or maintain a behaviour, an intention does not always translate into the desired behaviour [4]. This "*intention-behaviour gap*" can be attributed to self-regulation difficulties individuals experience [5, 6]. Self-monitoring is integral to self-regulation as it involves a conscious awareness of one's behaviour through systematic observation of goal-oriented behaviour [2]. Studies on behaviour change indicate consistent self-monitoring, regardless of the content or comprehensiveness of what is being monitored, is crucial for maintaining dietary changes over time and achieving goals [7–10].

While self-monitoring appears essential for bridging the "intention-behaviour gap", lack of adherence to dietary self-monitoring over time is often driven by current tools' complexity and time-consuming nature [11–13]. Paper-based tools (e.g., food journals or diet diaries) have been shown to decrease adherence to dietary self-monitoring within the first three to five weeks [12]. Technology-based tools, such as mobile applications, offer advantages over paper-based tools, including date and time stamps, instant feedback, and reminder signals to lessen the self-monitoring burden [14–16]. Although mobile applications have been found to be more effective than paper-based tools and websites in encouraging adherence to dietary self-monitoring [17, 18], inherent challenges are present with them that could reduce adherence [17–19]; for example, current mobile applications have an over-emphasis on calorie-counting and portion measurement [19]. In fact, trial data of a 24-week study shows a decline in adherence to dietary self-monitoring using mobile applications occurs during the fourth to ninth week [13]. Moreover, available applications are often dependent on numeracy and health literacy [20], meaning using these applications requires the ability to read, an adequate level of comprehension (i.e., the knowledge to itemize food items), math skills, and technological skills that may not be possible for everyone [21–23].

Shifting towards less complex and more user-friendly approaches, such as the plate-based approach to eating over itemizing foods, could enhance adherence to dietary self-monitoring [24]. The plate-based approach was developed in Sweden for patients with diabetes to teach meal planning which divides the serving plate into sections designated for specific foods. Using this approach for dietary self-monitoring includes filling in the proportions of each section to reflect the amount of each food type consumed. The use of the visual depiction of the plate, along with the lack of food itemization, texts, or numbers in the plate-based approach makes dietary self-monitoring less complex and more accessible, especially to individuals with varying levels of numeracy and literacy [20, 21, 25]. The plate-based approach to meal planning has been shown to improve health outcomes, with comparable efficacy to more

demanding interventions such as carbohydrate counting [20] or calorie counting [26]. Additionally, this approach has been associated with promoting healthy eating behaviours by reducing the consumption of refined carbohydrates, sugar, and total fat [26, 27].

In 2019, Canada joined a growing number of countries and associations [28, 29] in adopting the plate-based approach to healthy eating for the new national nutrition guideline [30]. This approach is embodied in the updated Canada Food Guide (CFG). It visually represents three main food groups on a "plate" (i.e., half the plate being vegetables and fruits, a quarter of the plate being protein, and the other quarter being whole-grain foods). The general message of CFG is to focus on the proportions of different food groups in one's diet rather than on serving sizes [30]. With no quantitative dietary recommendations, this simplified plate-based approach was recommended to be easier to comprehend and use for daily dietary planning [31]. Moreover, our previous study revealed a preference among adults for a plate-based dietary self-monitoring tool, as opposed to a traditional tool (e.g., a 3-day food diary), to make dietary changes over time [32].

This qualitative study aimed to explore the perceptions of both potential end-users (i.e., general public) and Registered Dietitians (RDs) of iCANPlate™, a mobile application that mirrors the CFG plate-based approach. RDs are considered the "knowledge-user" since they are the allied health professional who would integrate the application into their practice [33]. Results from this qualitative study will be used to further develop iCANPlate™ with the goal of helping Canadians make and maintain healthy dietary behaviours that are important to them. Including input by both "end-users" and "knowledge-users" at this stage of developing the application is critical and an important necessary first step; finding from this study will increase the likelihood of successful adoption, implementation, and continued use of the app over time [34, 35].

## Methods

### Study design

This qualitative study used online focus groups on Zoom with members of the general public and Registered Dietitians (RDs) and was carried out through the University of British Columbia (Vancouver, BC, Canada). Focus groups were conducted from July 2021 to August 2021 with an expected number of 45 participants per target audience [36, 37]. Our goal was to recruit sufficient participants to meet thematic saturation, defined as the point where no new themes or relevant information emerged through the thematic analysis [38]. This study followed the Consolidated Criteria for Reporting Qualitative Research (COREQ) [39] checklist (S1 Appendix).

### Participants and recruitment

Convenience sampling strategy was used to recruit members of the general public. we aimed to include a balanced representation of gender (i.e., man and woman) and diversity in age and cultural background. Recruitment methods included online advertisements on a research centre listserv at PERFORM Centre (Concordia University, Montreal, QC). Additionally, recruitment was carried out through the graduate student community listserv of the University of British Columbia and social media platforms such as Facebook and Twitter. Eligibility criteria for the general public included: English-speaking adults over 18 years familiar with operating a mobile device and had access to the technology required for video and audio connectivity (i.e., Zoom). We excluded individuals with cognitive impairment, a disability related to blindness or deafness, or those who did not know how to use a mobile application.

Based on convenience sampling, RDs were recruited via social media, primarily on Facebook, in various closed groups of dietitians across Canada. Eligible RDs were English-speaking (>18 years), familiar with mobile devices, and capable of participating in a focus group discussion over Zoom.

Interested participants from both groups contacted the research team via email. Researchers called participants to complete screening and determine eligibility. Following the screening process, eligible participants were informed about the study's purpose and procedure via a consent form on Qualtrics® (Provo, UT). Upon consenting, the same platform directed them to complete a sociodemographic questionnaire (age, gender, ethnicity). Members from the general public reported their highest level of education and previous experience with using dietary self-monitoring tools. RDs were asked to report their years of practice and experience in recommending self-monitoring tools during their clinical practice. After completing the survey, participants were scheduled to participate in one focus group at a convenient time.

## The dietary self-monitoring app: iCANPlate™

The original concept of iCANPlate™ was proposed by co-authors (TRC and JPG) to resemble the 2019 CFG. An application developer (Éphémère Creative Ltd) was hired to develop the prototype version 0.1. The iCANPlate™ application resembles a plate in the form of a pie chart whereby the end-user records meals and snacks by adjusting the proportions of each food group to match the proportions of the foods on their plate. The three food categories on the application match those recommended in the CFG. The app's main goal is to allow end-users to monitor foods consumed as they appear on a plate, categorizing the food into the appropriate CFG category and indicating the proportion of the plate occupied by the food. Importantly, iCANPlate™ is not intended to be used as a dietary assessment tool, as it does not ask users to record details or count serving sizes and calories.

## Focus group guide development

The Capability, Opportunity, Motivation, Behaviour (COM-B) model, as part of the Behaviour Change Wheel [40], was used to develop the focus group questions [41] since this model offers a comprehensive insight into the set of three factors that influence behaviour change. Table 1 shows the mapping of the focus group guide to the COM-B constructs. The focus group guide consisted of open-ended questions that covered three sections: Section 1- Perceptions of the 2019 CFG; Section 2- History of using dietary self-monitoring tools; Section 3- Content and features of the proposed dietary self-monitoring application (S2 Appendix). The same guide was used to facilitate all RD and general public focus groups. Following recommendations in a previous study [42], a pilot virtual focus group with six participants (non-dietitian students) was conducted to test and refine the guide to ensure its comprehensiveness [42]. Data from the pilot focus group were not analyzed or included in the results.

## Study procedure

Each focus group was facilitated by two trained female interviewers (one qualified RD (CCB) and a Nutrition graduate student (MKh)). A trained note-taker was available on each call to take notes and help with technical challenges. CCB had prior professional relationships with some RD participants. CCB was not involved in data coding to maintain impartiality in data analysis. Throughout the study, efforts were made to establish a good rapport with the participants.

Each focus group (4–9 individuals/focus group) started with an overview of the project, which included a brief description of the CFG for the general public focus groups and a

**Table 1. Mapping of the focus group guide to the COM-B constructs.**

| CAPABILITY | |
|---|---|
| Psychological capability | • What do you/ *your clients* know about Canada Food Guide?<br>• Which diet-tracking methods or applications have you ever used/ *suggested to your clients* (if any)?<br>• How do you suggest dairy products be tracked on the application?<br>• What other eating behaviours or elements of the CFG should be included in this application? |
| Physical skills | • What features are required to ensure accessibility for all users? |
| **OPPORTUNITY** | |
| Physical opportunity | • What makes it easy/hard for you/*your clients* to eat in accordance with the plate method that mirrors the CFG?<br>• Many "other foods" are not shown on the CFG. Which foods can you think of that you/ *your clients* would find difficult to represent on the plate? How do you suggest they be tracked on the app?<br>• How do you suggest beverages be tracked within the application?<br>• Which instructions should be provided to users to support their use of the app?<br>• What would be considered a successful day? |
| Social opportunity | • What features can facilitate social support and enhance user adherence to the application? |
| **MOTIVATION** | |
| Reflective motivation | • How do you view the application working on recording all meals throughout the day?<br>• Which features in the application could improve users' confidence when tracking their food intake?<br>• What makes it easy or hard for you/ *your clients* to use the tools you have experience with?<br>• What would be considered a successful day? |
| Automatic motivation | • What did/didn't you like about the plate-based app?<br>• Which features would incentivize users to adhere to using the app? |
| N/A | • Which other features can you think of that we did not discuss? |

*Italicized* words pertain to the Registered Dietitian focus groups only.

description of dietary self-monitoring for all focus groups. Participants then participated in a discussion based on Sections 1 and 2 of the focus group guide. Using Zoom's screen-sharing feature, participants viewed the prototype iCANPlate™ (version 0.1), followed by a discussion to explore their perceptions of the prototype. Participants did not have access to the application, nor were they required to download it. Following the completion of each focus group, the interviewer and note-taker conducted a debriefing session to determine the saturation point. Focus group sessions were audio-recorded, and participants were given appropriate honoraria to remunerate their time.

## Data analysis

After verbatim transcribing the audio recordings, transcripts were analyzed using the qualitative analysis software NVivo12 Pro (QSR International) (transcripts available in S4 and S5 Appendices). To ensure methodological integrity, we employed a combination strategy of inductive and deductive qualitative analysis. Following the thematic analysis outlined by Braun and Clarke [43], two independent researchers analyzed the transcripts from the RD group (MKh, RM) and the general public group (MKh, CB), to generate themes and subthemes. Data analysis involved an iterative process following coding techniques to identify themes and subthemes that emerged from the data. The language used for labelling the themes and subthemes was quoted directly from the participants' descriptive responses or the focus group guide. The first author (MKh) developed an initial codebook and a reflective report during the coding process (memoing) to demonstrate self-awareness in the analytical process and

enhance qualitative their skills. Both coders used the codebook for the final analytic coding. They reorganized it iteratively until the research team reached a consensus, ensuring a cohesive and agreed-upon approach to the analysis. While the identified themes were often linked to the specific questions in the guide, the participant-driven nature of focus group discussions and the open-ended questions allowed us to explore concepts not necessarily included in the focus group guide. Therefore, relevant themes were considered present even if unrelated to a particular question.

Two researchers (MKh and CB) then used deductive thematic analysis, whereby the COM-B model [41] was followed as a framework to organize the extracted subthemes. According to the available literature, the coders identified specific constructs that each subtheme pertained to. They regularly convened until they reached a unanimous consensus. While the interview guide was based on the COM-B model, the researchers analyzed the data in its literal form, meaning some of the participants' perceptions were coded into a different construct than initially intended. Throughout the analysis, discrepancies between coders were resolved via mutual discussions or consultation with a third independent reviewer (TRC). The latter was particularly beneficial when consensus could not be achieved through discussion alone.

## Ethical considerations

Ethics approval was obtained from the University of British Columbia (Behavioural Research Ethics Board Number: H21-01353] and Concordia University (Montreal, QC, Canada, Ethics Number: 30012869]. The study ensured confidentiality and anonymity of participants. Each participant was assigned a unique code to safeguard their identity, and a separate encrypted master list linked the codes to participant names. This password-protected file was securely stored and was accessible only by first author (MKh) and the corresponding author (TRC). During the focus group sessions, participants were encouraged to use nicknames for further identity protection. However, maintaining complete anonymity was challenging in the RD focus groups due to the potential recognition of participants from their work or school settings. Moreover, upon verbatim transcribing the audio recordings, any identifiable information associated with participants was removed.

## Results

In total, 72 individuals from the general public consented to the study, with 52 participating in nine focus groups (24 females, mean age of 40.2 (SD 18.4); 28 males, mean age of 42.6 (SD 16.5)); similarly, 56 Registered Dietitians consented, with 44 participating scheduled focus groups (39 females, mean age of 33.6 (SD 7.0); 5 males, mean age of 31.4 (SD 3.5)). The final sample size for summary statistics was smaller as some participants were not available for any of the focus groups or did not participate in their assigned focus groups and were unresponsive to follow-up attempts. Table 2 shows the socio-demographic characteristics of the study sample.

Focus groups were conducted from July 2021 to August 2021. The sessions ranged from 64 to 115 minutes (mean of 83 minutes). Thematic analyses revealed four main themes with 15 subthemes derived from participants. The main themes were clustered into two major categories, outlined in Table 3: the current iteration of iCANPlate[TM] described in the main text, and the components to include in future iterations of iCANPlate[TM] (available in S3 Appendix). Table 3 displays the major categories, themes and subthemes reported by participants as well as the mapping of each subtheme for both categories of main themes to a corresponding COM-B construct.

**Table 2. Socio-demographic characteristics of the study sample.**

|  | General Public | Registered Dietitian |
|---|---|---|
| Total number, n | 52 | 44 |
| Age (years), mean (SD) | 41.4 (17.2) | 33.4 (6.7) |
| Gender, n (%) |  |  |
| Man | 28 (54%) | 5 (11%) |
| Woman | 24 (46%) | 39 (89%) |
| Ethnicity, n (%) |  |  |
| White | 25 (48%) | 30 (68%) |
| Indigenous | 2 (4%) | 0 |
| South Asian | 2 (4%) | 1 (2%) |
| Chinese | 1 (2%) | 11 (25%) |
| Black | 1 (2%) | 0 |
| Filipino | 2 (4%) | 0 |
| Arab | 2 (4%) | 1 (2%) |
| West Asian | 9 (17%) | 0 |
| Others-mixed | 8 (15%) | 1 (2%) |
| Highest education levels, n (%) |  |  |
| Postsecondary certificate, diploma or degree | 46 (88%) | 44 |
| Some postsecondary education | 0 | 0 |
| Secondary (high) school diploma or equivalent | 6 (12%) | 0 |
| Less than secondary (high) school graduation | 0 | 0 |
| Years of practice, n (%) |  |  |
| ≤2 years | N/A | 14 (32%) |
| 3–5 years | N/A | 14 (32%) |
| 6–10 years | N/A | 11 (25%) |
| >10 years | N/A | 5 (11%) |
| History of using dietary self-monitoring tools, n (%) |  |  |
| Yes | 30 (58%) | 35 (79%) [a] |
| No | 22 (42%) | 9 (21%) [a] |
| Types of dietary self-monitoring tools used,[b] n (%) |  |  |
| Paper and pen food journals | 9 (17%) | 40 (32%) [a] |
| Itemizing mobile applications | 15 (29%) | 31 (25%) [a] |
| Web-based tools | 4 (7%) | 14 (11%) [a] |
| Simplified self-monitoring tools[c] | 7 (13%) | 24 (19%) [a] |
| Others | 3 (6%) | 3 (2%) [a] |

[a] RDs were asked for a history of recommending dietary self-monitoring tools to their clients

[b] Responses to this question (n = 30) were for those who had a history of using/recommending self-monitoring tools

[c] Checklists, portion counting, plate-based tool

N/A, Not applicable

## Theme 1- Facilitators to using the current iteration of iCANPlate™

**Subtheme 1- Self-awareness of dietary behaviours.** The general public and RD participants stated the app would foster self-awareness of overall dietary patterns rather than promoting calorie counting or itemizing specific foods. This aspect was perceived as an attractive feature of the plate-based approach, potentially increasing users' willingness to use the app. General public members explained general self-awareness of eating patterns, rather than needing to enter specific details, could facilitate achieving dietary goals.

Table 3. Major categories, themes and subthemes reported by participants.

| Categories | Themes | Sub-themes | Related COM-B construct |
|---|---|---|---|
| The current iteration of iCANPlate™ | A. Facilitators to use a plate-based dietary self-monitoring application | 1. Self-awareness of dietary behaviours | • Psychological Capability/ Reflective Motivation |
| | | 2. Simplicity | • Psychological Capability |
| | B. Barriers to using a plate-based dietary self-monitoring application | 3. Lack of food classifications | • Psychological Capability/ Physical Opportunity/ Reflective Motivation |
| | | 4. Conceptualizing proportions | • Physical Opportunity |
| | | Lack of inclusivity | • Social Opportunity |
| Components to include in future iterations of iCANPlate™ [a] | C. Essential components to add to iCANPlate™ | 5. Educational content and tutorials | • Psychological Capability/ Reflective Motivation |
| | | 6. Report dashboard | •. Psychological Capability |
| | | 7. Accessibility | • Physical Opportunity/ Capability |
| | D. Optional components to add to iCANPlate™ | 8. Personalization | • Reflective Motivation |
| | | 9. Automatic food logging | • Physical Capability |
| | | 10. Recording other eating behaviours | • Psychological Capability |
| | | 11. Social interaction | • Social Opportunity |
| | | 12. Professional support | • Physical Opportunity |
| | | 13. Interactivity | • Motivation/ Physical Opportunity |
| | | 14. Incentivization | • Reflective Motivation |

[a] Components to include in future iterations are outlined in S3 Appendix.

"*I think generally the concept is nice because it's more about patterns over the long term, and just to know what's on your plate, those proportions, versus not having to be super-exact with it every single time you log in. I think that's nice because it's easy to get sucked into numbers.*" [General Public- Focus group 2- Participant 2]

RDs stated the app would allow users to focus on improving their dietary behaviours, especially food quality, rather than promoting caloric restriction. RDs acknowledged the plate-based dietary self-monitoring was a weight-neutral approach focusing on diet quality rather than weight loss. They noted this approach differs from most publicly available dietary self-monitoring tools that focus exclusively on body weight and calorie counting, which could promote the development of eating disorders.

"*That's very cool because it's all about patterns, it's about your behavioral patterns and at the end of a period of time you get feedback on what you ate over that period of time, when you were eating it, how you were eating it.*" [RD- Focus group 2- Participant 3]

**Subtheme 2- Simplicity.** Focus group participants stated the simplicity of the app's usability would facilitate its use. They found the pie-like division of the plate to be familiar and straightforward. Participants noted the app's simplified interface did not require high literacy levels. A consensus emerged in all focus groups that data entry in the app was simple and would alleviate the annoyance of tedious logging, which is common in many dietary self-monitoring applications. Hence, they suggested the plate-based approach is more user-friendly and accessible to varying literacy and comprehension levels. One general public participant voiced:

"*First seeing the app, I did like the simplicity of it and maybe just like–it's not too simplistic but simple that everyone with different levels of literacy can use.*" [General Public- Focus group 6- Participant 2]

RDs highlighted the lack of quantification in iCANPlate™ for portions or calories, and how it would simplify the process of self-monitoring for their clients or patients. According to RDs, avoiding a number-focused approach can help their clients increase self-awareness of their dietary behaviours, especially those who struggle with understanding portions. RDs discussed by providing a simple method without the need to focus on numerical data, this plate-based approach could improve adherence to diet self-monitoring over time.

"*I think the visual with the different colours and just the proportion is very nice, and they can get rid of the numbers. I think it's a lot more user-friendly to most people.*" [RD- Focus group 3- Participant 4]

## Theme 2- Barriers to using the current iteration of iCANPlateTM

**Subtheme 3- Lack of food classifications.** RD and general public participants indicated that because several essential dietary components are not depicted in the literal picture of the 2019 CFG, users of iCANPlate™ might have difficulty using it. Specifically, general public participants suggested including other categories of foods that are depicted in the CFG, such as seasonings, sugary foods, cooking oil, and processed/ultra-processed foods. They also stated the app's inability to allow users to differentiate specific food qualities (e.g., low-fat versus high-fat food choices) would be a barrier to its usability.

"*To be honest it's selling a false pretense that you're eating vegetables and you're eating a healthy meal. If I buy Wendy's salad I know, oh my goodness. . . I should've just eaten a burger because I probably had more calories eating that salad.*" [General public- Focus group 1- Participant 2]

RDs voiced concern the lack of precision in classifying food qualities could create a misleading impression of an individual's diet, leading to inappropriate dietary recommendations for clients. An RD commented on the categorization of starchy vegetables in the fruits and vegetables section, highlighting the potential issue of providing advice that may not align with individuals' dietary goals and needs, particularly in cases such as diabetes. Another RD stressed the importance of including all foods and said:

"*I would say there's also the fact that people try to reproduce what's in the plate exactly. So, they're like, oh, but I'm eating excess food, but there's, it's not in the plate. So, can I eat it or not? So, it makes it difficult for them to know if you can include the food or not into their diet or not.*" [RD -Focus group 4- Participant 4]

There was also a consensus among the general public participants that recording all beverages in iCANPlate™ is important. RDs suggested creating separate beverage categories, such as hydrating/non-hydrating drinks, sweetened/unsweetened drinks, and alcohol. Furthermore, the absence of a separate category for dairy foods (milk and milk products) in the CFG might confuse future app users. They proposed emphasizing the dairy foods group either as a fourth separate group for logging or as a separate section in the application. RDs generally agreed, in alignment with the CFG, dairy products should be categorized under the protein category.

RDs proposed creating a distinct section dedicated to calcium or dividing the protein category into high-calcium and low-calcium protein sources to ensure adequate calcium consumption. An RD discussed the categorization of the dairy group under the protein category in the CFG, stating:

> "Being a plant-based dietitian for all my career is that there's so much misguidance right now that I'm happy there's not a dairy food group, but I'm not happy that there's not a calcium food group." [RD- Focus group 6- Participant 2]

**Subtheme 4- Conceptualizing proportions.**   The participants were concerned it would be harder to conceptualize the proportions of foods not typically served on a standard plate. Examples included foods that are not served on traditional dishes (e.g., sandwiches or smoothies), on dishes other than plates (e.g., bowls), or on varying plate sizes, or meals that are served on multiple containers and plates (e.g., multi-course meals). Participants also exemplified mixed dishes (e.g., lasagna) to be particularly challenging to conceptualize within the app unless the user made the dish or was familiar with the exact recipe. A member of the general public who had prior experience using the plate-based approach on a paper tool stated:

> "*I did this for a week on paper, and it works well for meals where you have a bunch of different things on the plate. But, if you're eating a stew or a bowl of pasta, it's harder to figure out because it's all in one pot.*" [General public- Focus group 2- Participant 1]

Given that this challenge was particularly attributed to the absence of serving or portion sizes, some general public participants suggested simple and universal methods to count portions (e.g., hand models or measuring cups) could be included as an optional feature. RDs discussed human error in estimating the proportions could have a negative impact on the accuracy of input data in the application. Thus, it is possible that the app's data would not provide reliable information to assess dietary behaviours. As RDs further emphasized, the iCANPlate™ application does not involve assessing the actual amount of each food group on the plate but rather their proportion relative to each other.

**Subtheme 5- Lack of inclusivity.**   Both groups mentioned the app might be less appropriate for individuals with specific health conditions or certain dietary requirements. A general public member living with diabetes stated the app might be less helpful for individuals with a particular health condition:

> "*I think the whole plate thing is geared for regular people, for people who have health problems or whatever, like for myself, I am diabetic, I don't know if I would enter my intakes.*" [General public -Focus group 4- Participant 2]

Conversely, some RDs suggested the plate-based approach in iCANPlate™ could benefit individuals with diabetes if certain modifications were made to the food groups outlined by the CFG. For example, an RD highlighted the importance of considering the carbohydrate content in fruits for individuals with diabetes and said:

> "*We should kind of tailor it a little bit. So rather than having both fruits and vegetables [together], it needs to be the veggie focus first.*" [RD- Focus group 4- Participant 4]

RDs agreed, in line with CFG recommendations, iCANPlate™ would not be appropriate for hospitalized patients or individuals with specific medical conditions, including cognitive

impairments or brain or spinal injuries. In addition, RDs with prior experience working with eating disorder populations noted recording dietary intake using the plate-based approach, similar to other dietary self-monitoring tools, could induce anxiety in this population.

In addition to the app's exclusion of people with diagnosed diseases, the lack of cultural inclusivity was also raised as a concern by both RDs and general public participants. The general public expressed recording some cultural foods within the app might be harder. The general public participants stressed if the goal of iCANPlate™ is to improve the dietary behaviours of Canadians, it should reflect the cultural diversities present in the country.

> "*I don't know if it [the app] would include all types of foods like different ethnic foods–for example, there are a lot of foods, fruits and vegetables from African or Chinese cultures, as well as Indigenous Canadian foods, so you have to think of that as well.*" [General public -focus group 4- participant 2]

Several RDs agreed individuals from diverse cultural backgrounds tend to consume mixed dishes. Yet, the application's compartmentalized nature made it difficult to conceptualize the various components of a mixed dish. Hence, they articulated the app may not reflect the average diet of their ethnically diverse clients. They also emphasized the crucial role of cultural inclusivity in dietary self-monitoring concerning acceptability and adherence to iCANPlate™ by end-users.

## Discussion

This qualitative study explored the perceptions of the general public and RDs of a plate-based dietary self-monitoring mobile application. The app's main goal is to self-monitor dietary intake with the idea of helping individuals bridge the "*intention-behaviour gap*" related to healthy eating behaviours. Our findings showed iCANPlate™ has the potential to facilitate these changes, given its simplicity and self-awareness approach. Yet, barriers to using the app were also discussed, particularly related to conceptualizing the proportions, lack of details on food classification, and inclusivity for cultural foods and health conditions.

Our study used the COM-B model as our framework to understand the barriers and facilitators to using the application from the intention-behaviour gap model. In this study, we report mainly on the capability and opportunity components, which is not surprising; given that self-monitoring is a post-intentional behaviour change technique [44], mainly used to increase *psychological capabilities* [45] and provide users with *opportunities* to incorporate healthy behaviours in their routines [46]. As a result, the *motivation* variables are commonly regarded as the precursors to intention formation [2, 47].

### Capability

Participants in this study agreed with the iCANPlate™'s focus on fostering self-awareness of dietary behaviours through self-monitoring proportions of different food groups on a plate rather than itemization. These findings align with another study conducted among breast cancer survivors living with overweight or obesity. According to their participants, using a dietary self-monitoring application improved their self-awareness, resulting in lower calorie intake by preventing mindless calorie consumption [48]. Moreover, the app's ability to increase self-awareness of dietary behaviours was particularly valued over apps that track weight or calories. As reported in a qualitative study, weight management app users generally disliked applications that emphasize calorie counting rather than motivational methods to modify eating

behaviours [19, 49, 50]. Calorie counting is excessively complicated and associated with the development of restrained eating [19, 51, 52], thereby may reduce adherence.

The simplicity of the interface of iCANPlate[TM] was identified as another facilitator for its use and a way to improve adherence to dietary self-monitoring. Participants agreed the simplified food/drink logging method would make the plate-based dietary self-monitoring app accessible and visually straightforward to varying levels of literacy and numeracy skills. This could be explained by the reported association between the accuracy of portion-size estimation and literacy and numeracy skills [21]. Therefore, the absence of portions and serving sizes in iCANPlate[TM] has the potential to make nutrition information more accessible for individuals with limited literacy and numeracy skills. The finding from this study is consistent with a systematic review [53] that highlights the importance of ease of use and minimal input in promoting engagement with health and well-being mobile applications. Supporting our findings, that systematic review further noted these factors are associated with low cognitive load [53].

## Opportunity

User-friendliness, as part of the application's simplicity, was identified under the opportunity component influencing eHealth utilization [54]. Our findings regarding the simplicity of the plate-based approach are consistent with another study [20] which evaluated the effectiveness of the plate-based approach for nutrition education in a population with diabetes (n = 150, median age (IQR) = 55 (45, 60)). In that study, the plate-based approach to nutrition education improved health outcomes among participants with low numeracy skills compared to the carbohydrate counting method of other dietary self-monitoring tools [20]. However, our participants expressed the app's simplicity would render it difficult to log foods that were not explicitly depicted in the visual representation of the CFG. Concerns were also raised about logging beverages. Given the high prevalence of sugary drinks and sugar-sweetened beverages consumption (23% of mean energy intake) [55], it is crucial to include the recording of all beverages in dietary self-monitoring.

Our study's findings suggest foods from different cultures may be more challenging to record within the current iteration of iCANPlate[TM]. As a limitation of the 2019 CFG, other qualitative research on parents has also shown the 2019 CFG does not reflect their traditional foods [56]. In the absence of cultural food representations on the CFG, incorrect assumptions can be made regarding cultural foods' health benefits and nutritional values [57]. Our findings are in line with the results from a multi-lingual survey-based study where the general public members and health care professionals (total n = 2,382) agreed missing major food items (e.g., local foods) in digital health interventions are significant barriers to selecting and engaging with a dietary self-monitoring tool [52]. This finding was confirmed with another qualitative study whereby Saudi women with overweight and obesity expressed the need for culturally sensitive information in an ideal weight management application [58]. In the Canadian context, up to 40% of the population will be represented by visible minorities by 2036 [59]; therefore highlighting the significance of incorporating cultural considerations in the future versions of iCANPlate[TM].

## Motivation

Although the simplicity of iCANPlate[TM] could improve its user-friendliness, concerns were expressed about its inadequate classifications, particularly in relation to food quality. These concerns were similar to the Helsel et al. study that reported a simplified method for dietary self-monitoring can lead to similar short-term weight loss compared to detailed dietary self-monitoring; the simple approach used check marks in boxes to estimate the fat content and

portion size (based on energy content) of meals and snacks [24]. It will therefore be important to include certain aspects of diet quality in a simplified dietary self-monitoring tool (i.e., iCAN-Plate[TM]) so as to improve health outcomes [60]. Further research is required to determine how to best represent these categories of foods within iCANPlate[TM] while also maintaining its simplicity.

## Strengths and limitations

A strength of this study was our ability to recruit an equal ratio of males and females from different ethnic backgrounds to participate in the general public focus groups. This helps to represent the diverse population living in Canada and can increase the reliability of the collected information [61]. Moreover, we engaged both members of the general public and RDs. Involving potential end-users and RDs in the development of iCANPlate[TM] is particularly important since it will increase the chance of obtaining higher levels of appreciation and prolonged use of the application by both the end-users and knowledge-users [34].

Nevertheless, this study is not without limitations. First, RD participants were primarily females of white ethnicity, reflecting the profession's composition in Canada. However, most RDs reported having clients from different ethnic backgrounds whose needs were incorporated into the focus group discussions and data analysis. Secondly, the general public participants in this study had a comparatively high level of education. Although the health literacy levels of general public participants were not measured, they demonstrated self-awareness about their health and dietary behaviours during the focus group discussions. Given the known association between education and health literacy [62, 63], these factors may have influenced the study's results. Exploring the perceptions of individuals with varying levels of education and health literacy skills regarding the use of iCANPlate[TM] is crucial to investigate the hypothesis that the application's simplicity will facilitate the process of dietary self-monitoring for the general public across socioeconomic groups.

## Conclusions

This qualitative study explored the perceptions of potential end-users and Registered Dietitians of the prototype iCANPlate[TM], a dietary self-monitoring tool based on the plate-based approach. Participants appreciated the increased self-awareness of dietary behaviours brought forth by iCANPlate[TM], whereby the application does not focus on numbers or serving sizes but on how food is proportioned on the plate. The simplicity of the application was discussed as a principal factor in making iCANPlate[TM] accessible to individuals with varying literacy levels. Findings for future iterations of the app included a range of suggested features to improve the added value of this dietary self-monitoring tool. Exploring the perceptions of individuals from lower levels of education and health literacy are needed in future to ensure the accessibility of iCANPlate[TM] for this population.

## Supporting information

**S1 Appendix. Consolidated Criteria for Reporting Qualitative Studies (COREQ): A 32-item checklist.**
(DOCX)

**S2 Appendix. Focus group guide.**
(DOCX)

**S3 Appendix. Components to include in future iterations of iCANPlate[TM].**
(DOCX)

**S4 Appendix. General public focus group transcripts.**
(ZIP)

**S5 Appendix. RD focus group transcripts.**
(ZIP)

## Acknowledgments

The authors appreciate the contribution of all study participants and the research team, particularly Patricia Angeles and Trista Yue Yuan, for helping the study team conduct the focus group discussions. The authors are also grateful to the Nutrition and Eating Behaviour Lab members at the University of British Columbia for their support and expertise, which played a significant role in completing this study.

## Author Contributions

**Conceptualization:** Maryam Kheirmandparizi, Ryan E. Rhodes, Tamara R. Cohen.

**Data curation:** Maryam Kheirmandparizi.

**Formal analysis:** Maryam Kheirmandparizi, Coralie Bergeron, Rana Madani Civi.

**Funding acquisition:** Jean-Philippe Gouin, Maryam Kebbe, Ryan E. Rhodes, Biagina-Carla Farnesi, Nizar Bouguila, Tamara R. Cohen.

**Investigation:** Maryam Kheirmandparizi, Jean-Philippe Gouin, Celeste C. Bouchaud.

**Methodology:** Maryam Kheirmandparizi, Jean-Philippe Gouin, Celeste C. Bouchaud, Maryam Kebbe, Nizar Bouguila, Tamara R. Cohen.

**Project administration:** Maryam Kheirmandparizi, Celeste C. Bouchaud.

**Resources:** Maryam Kebbe, Tamara R. Cohen.

**Supervision:** Ryan E. Rhodes, Annalijn I. Conklin, Scott A. Lear, Tamara R. Cohen.

**Validation:** Tamara R. Cohen.

**Writing – original draft:** Maryam Kheirmandparizi.

**Writing – review & editing:** Jean-Philippe Gouin, Celeste C. Bouchaud, Maryam Kebbe, Ryan E. Rhodes, Biagina-Carla Farnesi, Annalijn I. Conklin, Scott A. Lear, Tamara R. Cohen.

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
