## [Decision Letter · Decision Letter 0]

27 Sep 2023

PONE-D-23-17392Perceptions of self-monitoring dietary intake according to a plate-based approach: A qualitative studyPLOS ONE

Dear Dr. Cohen,

Thank you for submitting your manuscript to PLOS ONE. After careful consideration, we feel that it has merit but does not fully meet PLOS ONE’s publication criteria as it currently stands. Therefore, we invite you to submit a revised version of the manuscript that addresses the points raised during the review process.

We look forward to receiving your revised manuscript.

Kind regards,

Sana Sadiq Sheikh

Academic Editor

PLOS ONE

3. We note that Figure 1 in your submission contain copyrighted images. All PLOS content is published under the Creative Commons Attribution License (CC BY 4.0), which means that the manuscript, images, and Supporting Information files will be freely available online, and any third party is permitted to access, download, copy, distribute, and use these materials in any way, even commercially, with proper attribution. For more information, see our copyright guidelines: http://journals.plos.org/plosone/s/licenses-and-copyright.

5.Please review your reference list to ensure that it is complete and correct. If you have cited papers that have been retracted, please include the rationale for doing so in the manuscript text, or remove these references and replace them with relevant current references. Any changes to the reference list should be mentioned in the rebuttal letter that accompanies your revised manuscript. If you need to cite a retracted article, indicate the article’s retracted status in the References list and also include a citation and full reference for the retraction notice.

Additional Editor Comments:

Reviewer 1 comment:

The authors should highlight more on how you one can extrapolate the result to many multi-cultural & ethnic population which is increasing in Canada day by day due to influx of immigrants. In addition this would have increased the usefulness of this app in other countries as well.

Reviewers' comments:

Reviewer's Responses to Questions

**Comments to the Author**

1. Is the manuscript technically sound, and do the data support the conclusions?

Reviewer #1: Yes

Reviewer #2: Yes

2. Has the statistical analysis been performed appropriately and rigorously? 

Reviewer #1: Yes

Reviewer #2: N/A

3. Have the authors made all data underlying the findings in their manuscript fully available?

Reviewer #1: No

Reviewer #2: No

4. Is the manuscript presented in an intelligible fashion and written in standard English?

Reviewer #1: Yes

Reviewer #2: Yes

5. Review Comments to the Author

Reviewer #1: Thank you for considering me as a reviewer for this paper. I thoroughly enjoyed reading this paper.

The article intents to explore the use of a mobile app to explore dietary self-monitoring using Plate-based approach, since Canadian recent dietary guidelines are based on this concept with 3 sections on a plate. This is a pertinent and relevant topic as most tools to assess dietary intake are very extensive, laborious, requires higher literacy levels among respondents and a lot of number recording which is often boring plus it can be self-esteem damaging and does fulfil the purpose of dietary monitoring to improve health.

In general, this is a potentially stimulating qualitative research manuscript in the area of nutritional sciences which is very much needed to give relevance to the findings in quantitative research. Presumably the authors and their team in the past have done quantitative work on this app development process. The manuscript has been well written with clarity and detailing in methodology. It is encouraging to see that the authors have followed COREQ guidelines in reporting qualitative work which further authenticates the manuscript and give relevance to qualitative research in the area of diet and nutrition. The results are encouraging, however more quotes from both the dietitians and the general public could have added the qualitative strength to this article.

Given that the aim of the study was also to explore the efficacy of the app among all major ethnic groups within Canada. I think the authors should highlight more on how you one can extrapolate the result to many multi-cultural & ethnic population which is increasing in Canada day by day due to influx of immigrants. In addition this would have increased the usefulness of this app in other countries as well.

Reviewer #2: Review of ‘Perceptions of self-monitoring dietary intake according to a plate-based approach: A qualitative study’

I have reviewed the research paper ‘Perceptions of self-monitoring dietary intake according to a plate-based approach’. First off: maybe I missed this information, but why are certain sentences blacked out? Why aren’t reviewers allowed to see who gave the ethical approval (line 231), for instance? I cannot sign off the ethical approval for this reason (again, for instance). A second not entirely substantive point: I would not consider myself an expert on qualitative research. My review will therefore be less in-depth than perhaps desirable.

The paper describes the qualitative research of a dietary self-monitoring tool, called iCANPlate, on two population samples, i.e., general population and experts on the topic, in focus groups. In an intelligible manner, the authors cover the participants’ assessment of the tool on the basis of three themes, i.e., capability, opportunity and motivation. The conclusions drawn come down to the assertion that the simplicity of the tool can help people change their dietary behaviors, albeit with the downside of lacking cultural diversity. An important flaw is constituted by the use of convenience sampling, which may be the reason that the samples seem to lack representativity. However, this is recognized by the authors.

My (short) conclusion would be that this paper is well-written and serves its purpose, as a qualitative explorative study. I would think that as such, it most likely can be considered as a precursor to a more in-depth study into the qualities and usefulness of the tool iCANPlate.

6. PLOS authors have the option to publish the peer review history of their article (what does this mean?). If published, this will include your full peer review and any attached files.

Reviewer #1: No

Reviewer #2: No

---

## [Author Response · Author response to Decision Letter 0]

2 Nov 2023

Response 1: Thank you for supplying the templates, these changes have been made to the title page: Author titles removed (see lines 4-6), unnecessary information (full addresses) removed from affiliations (see lines 8-19), affiliations have also been reorganized to match preferred structure (also lines 8-19), and physical address removed from corresponding author section and the initial are added (see lines 20-21). 

As for changes to the main body, they are: correction of all levels of heading to correct font size (see lines 23, 53, 63, 121, 122, 130, 154, 165, 179, 194, 221, 233, 257, 258, 278, 298, 299, 334, 354, 387, 459, 479, 491, 497, 504, 506, 511), removal of brackets on table citations (see lines 169, 238, 251), change of brackets from round to square on all intext citations (see lines across entire document for changes), and the removal of funding from acknowledgements (see line 492). 

Response 2: Thank you for your query. We have taken the necessary steps to make the transcripts of the focus groups available as Supporting Information files. Please find the transcripts as S4 Appendix. General public focus group transcripts and S5 Appendix. RD focus group transcripts in the supporting information list accompanying the manuscript. DOI will be available after acceptance.

Response 3: The figure has been removed from the submission as it holds minor relevance and was not necessary to get the sections point across. Lines 154-164 indicate changes in wording to reflect the removal of Figure 1 from the manuscript. 

Response 4: Thank you for notifying us about this requirement. We have incorporated captions for the Supporting Information files at the end of our manuscript (Lines 692-698). Additionally, we've ensured that all in-text citations are now matched accordingly.

Response 5: Thanks for bringing this to our attention. After carefully reviewing the reference list, we identified an issue related to the reference management software we were using, which inadvertently affected the accuracy of the references listed in the initial submission (references #17-21). We have now rectified this issue, and the reference list has been updated to ensure that all references are correct and up to date. All references are now listed according to the PLOS ONE requirements. More details are listed below:

• All journal name abbreviations have been included and the reference style is according to the Vancouver style.

• For all in-text citations, the reference number is now in square brackets.

• Removed references that were not cited in the text (#19, 20, 21) and replaced with the correct references.

• References #17,18 are now replaced with the correct references.

• Reference #52 initially located inaccurately, is now correctly attributed to its respective sources within the reference list as #19.

• Removed reference #27 as I could not find it in journal databases (probably a retracted paper)

• We have recently published a paper on the paper-based version of the plate method which we cited in the text and included in the reference list (#32).

• Based on the comments by reviewers to emphasize the potential applicability of our research findings to a diverse range of multicultural and ethnic populations, particularly within the context of Canada's increasing immigrant population we added another paper to the text and included it in the reference list (#59). 

Additional Editor Comments

Response 3: Thank you for the comment. We have made our data (including all focus group transcripts) available now. 

Response 5: Thank you for the comment. We have added additional quotes to the paper.

Reviewer 1 comment- 

Response: Thank you for your feedback. We have addressed your suggestion by including the following sentence in the manuscript (reference #59): "In the Canadian context, up to 40% of the population will be represented by visible minorities by 2036 [59]; therefore highlighting the significance of incorporating cultural considerations in the future versions of iCANPlateTM." This addition underscores the growing multicultural and ethnic diversity in Canada and the importance of considering these factors in future iterations of the iCANPlateTM app, making it more applicable not only in Canada but also in other countries with similar demographic trends.

Reviewer 2 comment- 

Response: Thank you for reviewing our paper. We appreciate your feedback and apologize for any confusion regarding the blacked-out information. To avoid any type of biases and to adhere to ethical guidelines, we initially redacted certain identifiable information in the manuscript. However, in response to your concern and to ensure transparency, we have now made this information visible. You will find the necessary details regarding the ethical approval on line 224-225 and any other relevant information. We hope this addresses your concern, and we are grateful for your attention to this matter.

---

## [Editor Report · Decision Letter 1]

7 Nov 2023

Perceptions of self-monitoring dietary intake according to a plate-based approach: A qualitative study

PONE-D-23-17392R1

Dear Dr. Cohen,

We’re pleased to inform you that your manuscript has been judged scientifically suitable for publication and will be formally accepted for publication once it meets all outstanding technical requirements.

Kind regards,

Sana Sadiq Sheikh

Academic Editor

PLOS ONE

---

## [Editor Report · Acceptance letter]

15 Nov 2023

PONE-D-23-17392R1 

Perceptions of self-monitoring dietary intake according to a plate-based approach: A qualitative study 

Dear Dr. Cohen:

I'm pleased to inform you that your manuscript has been deemed suitable for publication in PLOS ONE. Congratulations! Your manuscript is now with our production department. 

Kind regards, 

on behalf of

Dr. Sana Sadiq Sheikh 

Academic Editor

PLOS ONE